# Psychological capital and social class: A capital approach to understanding positive psychological states and their role in explaining social inequalities

Deborah De Moortel[1,2]*, Mattias Vos[1], Bram Spruyt[1], Christophe Vanroelen[1], Joeri Hofmans[3], Edina Dóci[4]

1 Department of Sociology, Brussels Institute for Social and Population Studies, Vrije Universiteit Brussel, Brussels, Belgium, 2 Flanders Research Foundation, Brussels, Belgium, 3 Department of Psychology, Work and Organizational Psychology, Vrije Universiteit Brussel, Brussels, Belgium, 4 Louvain Research Institute in Management and Organizations, Louvain School of Management, Université Catholique de Louvain, Louvain-la-Neuve, Belgium

* deborah.de.moortel@vub.be

## Abstract

Psychological capital (PsyCap) is a multidimensional concept entailing hope, self-efficacy, optimism, and resilience. This paper argues that it can be considered a form of "capital" explaining social inequality. We test whether PsyCap can be integrated into the Bourdieusian capital framework by assessing its relationship with social, economic, and cultural capital. We also identify different types of social positions based on the volume and composition of psychological, economic, cultural, and social capital. We use cross-sectional data from the European Social Survey of 2012 (N = 35,313 respondents; 29 countries). To test the associations with the Bourdieusian capital types, we calculated multilevel spearman rank correlations and performed confirmatory factor analyses (CFA). Latent Class Analysis identified different types of social positions. We found positive weak correlations between PsyCap and the indicators of cultural capital ($r \leq .14$) and positive moderate correlations with the indicators of economic and social capital ($r \leq .24$). The results of the CFA showed that the fit of the 4-capital model was superior to that of the 3-capital model. We identified six types of social positions: two deprived types (with overall low capital levels); two well-off types (with overall high capital levels) and two types with high psychological and social capital in combination with varying levels of cultural and economic capital. Including PsyCap in the Bourdieusian capital framework acknowledges the power of positive psychological states regarding processes of social mobility and social inequality on the one hand and calls for understanding PsyCap as a social and group-level phenomenon on the other hand. As such, integrating PsyCap into the Bourdieusian framework can help to address the long-standing issue of understanding the relationship between social and individual differences in the study of social inequalities.

**Data Availability Statement:** The data underlying the results presented in the study are available at https://ess-search.nsd.no/.

**Funding:** This research is facilitated by the research grant 'FWO1.2.T82.21N', which is assigned to the first author by the Research Foundation Flanders. The funders had no role in study design, data collection and analysis, decision to publish, or preparation of the manuscript.

# Introduction

Social inequality is on the rise [1]. More and more people have few possibilities to earn a decent living, or to have a fair chance at a good life [2]. Pierre Bourdieu inspired many scholars to think about social inequality as a result of combinations of economic, cultural, and social capital. However, recent theoretical work called to integrate the notion of psychological capital (PsyCap), a concept originating from positive psychology, into Bourdieu's capital framework [3, 4]. PsyCap is defined as an individual's positive psychological state of development, characterized by the psychological resources of (1) self-efficacy (i.e., having the confidence to take on challenging tasks), (2) optimism (i.e., a positive attribution about the future), (3) hope (i.e., persevering towards a goal), and (4) resilience (i.e., bouncing back when having set-backs) [5].

Including PsyCap in the Bourdieusian capital framework acknowledges the power of positive psychological states regarding processes of social mobility and social inequality [4]. Positive psychological states can, just like the Bourdieusian capitals, lead to differences in access to resourceful connections [6], good jobs [7], wealth [8], and high-quality life [9]. Yet, this has largely been ignored in sociological theories that try to explain the drivers of social inequality, like the theory of Pierre Bourdieu [4]. Although, PsyCap has similar qualities as Bourdieu's capitals, it is distinct from them because PsyCap is, unlike the forms of capital identified by Bourdieu, partly independent from socialization. Up until now, however, these arguments are voiced in theoretical articles while empirical tests of these ideas are still lacking. Addressing this issue, the first aim of this study is to assess whether PsyCap can be empirically integrated into the Bourdieusian capital framework.

Moreover, according to Bourdieu's social class framework, every form of power or resource in social life (such as capital) is a criterion for distinguishing social positions [10, 11]. Bourdieu explains that the closer one individual is to another individual based on their possession and utilization of various capitals in the social space, the more likely these similarly located individuals are to have the same social position [12]. Because PsyCap, just like economic, social, and cultural capital can be deemed a resource in life and a principle of distinction, we believe it can also be used to define positions in the social space. Therefore, our second aim is to find different types of social positions based on the volume and composition of psychological, economic, cultural, and social capital.

To address these aims, we utilize a transnational approach [13]: we examine the relations of PsyCap with the Bourdieusian capitals and the belonging of individuals to different types of social positions in Europe. Using a transnational approach helps to develop a broadly applicable social class framework. Research comparing social classes across countries, or social class research within a specific country usually adopt an established social class framework that is applicable broadly, such as the Erikson-Goldthorpe-Portocarero (EGP) social class scheme [14] or (neo-)Marxist social class schemes [15]. In that sense, the types of social classes that can be found within the population are assumed to be valid over different high-income countries.

Yet, despite that social class frameworks are applicable in a pan-European context, countries exhibit different degrees of income inequality, varying levels of educational democratization and different welfare regimes [16], which affect the sizes of the classes. Therefore, our final aim is to examine country differences in the distribution of social classes.

We first discuss Bourdieu's capital approach to explaining social inequalities and review his cultural reproduction theory to understand the origins of social classes. Then, we show that his theory misses the potential of psychological resources to explaining social inequalities. Subsequently, using the European Social Survey (ESS), we analyze how PsyCap is relatively independent from, but also related to other forms of Bourdieusian capital. Then, we present results from latent clustering analysis (LCA) to describe the underlying class structure in Europe

based on the four types of capital. We conclude the results section by presenting country-specific distributions of the "integrated" class structure. The paper ends with a discussion on the implications of our results.

## Bourdieu's capital approach

Bourdieu's capital approach is an attempt to unite the micro- and macro-perspective to understanding social inequality. Capital, as defined by Bourdieu, refers to the whole of means that can be used to effectively defend one's place in society and possibly improve one's position in relation to others [10, 11]. On the one hand, Bourdieu underscores the micro-perspective by stressing the importance of individual-level differences in *volume* and *composition* of economic, cultural, and social capital for explaining social inequality [11]. Economic capital refers to the resources of individuals which are directly exchangeable for money. Social capital refers to the relations and memberships someone can use to reach their goals. Cultural capital is a configuration of dispositions, behaviors, and accomplishments in cultural practices and value orientations. There are three kinds of cultural capital. Firstly, cultural capital can be embodied. This means that experiences of upbringing will be internalized as lasting tastes and dispositions. This type of cultural capital is closely related to the concept of "habitus". Tastes, preferences, and certain kinds of (practical) knowledge may become a person's 'second nature' over time. Secondly, cultural capital can be objectified, such as books, paintings, machines, and instruments. Thirdly, cultural capital can be institutionalized, such as titles and diplomas [11]. Research shows that people's capital volume and composition are clearly related to a variety of outcomes in life such as job status, and even physical health [12].

On the other hand, Bourdieu's theory of cultural reproduction also explains how capital volume and composition is dependent upon the societal structure (i.e., the macro-perspective). It is especially the concept of habitus that links the individual to the social [17]. He explains that the lower classes also have culture, but that their cultural background is of less value to succeed in life. This is because the dominant groups (i.e., the elite) determine what cultural background is valuable to succeed and because the cultural background of the dominant group is permeated in the institutions [17]. With institutions (e.g., educational systems; labor markets) mostly adapted to the culture of the dominant groups, individuals with a more valued cultural background (thereby higher "cultural capital") can become more successful than individuals with a less valued cultural background (rendering them to have low cultural capital). In short, Bourdieu was able to introduce the cultural elements in highly deterministic macro-theories and is therefore known for attempting to incorporate human agency into structural theories of social inequality [18].

The notion of the different forms of capital draws attention to the transformation process, whereby one type of capital is used to acquire another. All types of capital can be transformed into another, but this typically takes time and/or effort and some types of capital (e.g., economic capital) are more easily transformed than others. Consequently, two individuals with the same level of one type of capital (e.g., economic capital) can lead totally different lives depending on their time and effort to acquire other types of capital. For instance, a person can invest money (economic capital) to get access to higher education (cultural capital). Each capital can be deployed for investments increasing wealth, knowledge, or the social network and thus ultimately to create more capital. This draws attention to the *composition* of capital (as different from the volume of capital) that people possess [10]. Based on the composition and volume of capital, Bourdieu defines different fractions in the social strata. For instance, among the middle class there is a fraction who are high in cultural and low in economic capital on the one hand and a fraction who are high in economic and low in cultural capital on the other hand.

In addition, the notion of the different forms of capital draws attention to the intergenerational transmission process, whereby parents transmit their social position to their children. Here too, both the volume and the composition are important. For instance, in general the middle class is more dependent on cultural capital (such as education), than on economic capital for inter-generational transmission of their social position (and the associated economic capital) [19]. There is transmission of resources across generations, through inheritance of material resources and through learning environments (especially schools). So, people come into educational systems with very different "starting" capitals. As a result, inequality increases in the long run.

In sum, people can collect more capital and different types of capital via their own deliberate efforts and via intergenerational transmission. Bourdieu uses the different forms of capital to highlight the fact that there are transformation and transmission processes between the different forms of capital (in all directions). Formulated differently, capital is a set of resources that can be accumulated or depleted over a lifetime and/or over generations. Therefore, capital is also referred to as accumulated history [11].

## Positive psychological states as a form of capital

Although Bourdieu paid attention to individual differences when explaining social inequality, he neglected the role of individual dispositions, such as people's differing tendencies to experience positive psychological states (PsyCap). Yet, this can play an important role in explaining social (im)mobility on the social ladder. For example, due to different dispositions and abilities, two children from the same family—having similar economic, social and cultural capital—can have very different educational outcomes [20]. Bourdieu's approach falls short on this occasion, because the home environment (i.e., capital transmission) cannot fully explain the differential educational outcomes. While Bourdieu's capitals are, by definition, a hundred percent socialized, PsyCap has a trait-based component [4]. This might better explain why some children have more chances to succeed, even though everything in their home environment worked against their success. These observations have often been treated as "marginal" and are therefore greatly undertheorized [4]. In Bourdieusian-inspired work, psychological states have often been treated as something that has to be explained, rather than something that has independent explanatory power [21].

However, PsyCap has a "nurture" component as well; through social input, learning and development, PsyCap can grow or decline [5]. Dóci et al. [3] drew attention to social inequalities in people's access to PsyCap. They postulate that not everyone has the same chances for developing high PsyCap, but that it depends strongly on one's social position. Dominant social groups will have a better chance to develop high PsyCap for various reasons. Firstly, these groups tend to be more positively perceived by their environment. Consequently, their repeated "success" experiences and the social confirmations they receive make it easier for them to generate positive PsyCap. Secondly, members of dominant social groups get more opportunities to prove their mastery, resulting in higher chances to increase their PsyCap. Thirdly, high-power individuals get more positive responses to their agentic and goal-striving behaviors, and they also have more access to differential pathways towards achieving their goals and attaining success [3].

## Objectives

PsyCap can thus be considered a form of capital in the Bourdieusian sense. PsyCap fits nicely the interpretation of (1) "investment" (PsyCap translates into positive career outcomes or social status), (2) a mechanism of differentiation and unequal distribution (see above), (3)

"accumulated history" (PsyCap can be transformed into other types of capital and intergenerationally transmitted), and (4) it cannot be confined to any other type of capital (because it is partly independent from socialization, for an elaborate theoretical argumentation see, Dóci et al. [4]). We empirically test the idea of PsyCap being a separate form of capital (Objective 1). On the one hand, to distinguish a form of capital, it needs to be related (correlated) to the other types of capital, because relations of transmission and transformation should be possible. On the other hand, PsyCap also needs to be relatively independent from other forms of capital (no substantial correlations), otherwise it cannot be seen as a different capital type. Based on the theoretical considerations outlined above, we hypothesize that PsyCap should be related to yet independent from the other types of capital.

In addition, based on the capital volume (i.e., having more or less) and composition (i.e., more or less of different types), fractions in the social strata can be defined. Previous research, using a transnational approach in Europe, has found groups with high economic capital (managers, professionals, entrepreneurs and executives), but varying levels of cultural capital, and groups with low economic capital (routine employees, industrial workers) having both high and low levels of cultural capital [22]. Another transnational study in Europe [13], found three groups. One with high levels and one with low levels on all the Bourdieusian capitals and, finally, a group with high informal social capital, but varying levels of economic and cultural capital. These results fit nicely with Bourdieu's observation that there are meaningful class fractions within classes [17]. By including PsyCap into his social class framework, we expect to find a dominant class with high levels of Bourdieusian capitals, and high levels of PsyCap. Because repeated experiences of success affect PsyCap positively, we expect a positive relation between membership to a dominant group and high PsyCap. On the contrary, because repeated experiences of failure affect PsyCap negatively [23], we expect to find a group with low levels on all the forms of capital (including PsyCap). In addition, we expect to find fractions within the middle class with varying levels on the four types of capital. We set out to reconstruct the social space using all four forms of capital (Objective 2).

Finally, our last aim (Objective 3) is to examine cross-national differences in social classes distribution. The sizes of social classes within European countries are influenced by macro-level factors, such as a country's level of economic development, income inequality and educational expansion [24, 25]. Countries with less economic development and high-income countries with large income inequality demonstrate larger (self-identified) low social classes [25]. Moreover, educational expansion can lead to large groups of high classes (with high volume of cultural and economic capital) [24]. Because there are transformation processes between all types of capital and because a country's macro-characteristics influence capital volume and composition, it is hypothesized that there will be a greater proportion of social classes with low general capital volume and deprived capital composition in countries with low economic development (such as Ukraine, Kosovo and Albania), in economically developed countries with high income inequality (such as Israel, Southern European and Anglo-Saxon countries) and in countries with less developed educational expansion (such as Southern European countries), while a greater proportion of social classes with high general capital volume and rich capital composition will be found in their counterparts (such as Western and Northern European countries).

## Materials and methods

### Data

This study used data of the ESS, which is a biennial cross-national survey in Europe, conducted since 2001 (https://www.europeansocialsurvey.org), with changing thematic modules each

year and collected via face-to-face interviews. Round 6, with participants recruited in 2012 and 2013, was used because this wave has the most recent module on personal and social well-being. The ESS6 module sought to incorporate a new validated scale of positive well-being and includes questions on well-being promoting behaviors. Therefore, ESS6 has the most recent and comprehensive scale on psychological capital. The ESS6 includes representative samples of persons aged 15 and over, who are resident in one of 29 European countries (Albania, Belgium, Bulgaria, Switzerland, Cyprus, Czechia, Germany, Denmark, Estonia, Spain, Finland, France, United Kingdom, Hungary, Ireland, Israel, Iceland, Italy, Lithuania, Netherlands, Norway, Poland, Portugal, Russian Federation, Sweden, Slovenia, Slovakia, Ukraine, Kosovo). All countries were included in the analyses. Respondents younger than 21 and older than 65 were excluded from the analyses, to ensure a relatively homogeneous group with similar chances to have completed education and to have a decent financial situation. As the current study utilized secondary data it was exempt from the ethical review process of the Ethics Committee Human Sciences of the Vrije Universiteit Brussel. The authors had no access to information that could identify individual participants during or after data collection.

### Variables

**Psychological capital.**    There are 6 questions in the ESS6 which can be used as indicators of a proxy measure of PsyCap: (a) At times I feel as if I am a failure, (b) In general I feel very positive about myself, (c) I am always optimistic about my future, (d) There are lots of things I feel I am good at, (e) When things go wrong in my life it takes a long time to get back to normal, and (f) How difficult or easy do you find it to deal with important problems that come up in your life. For the latter, answer categories ranged from 0 (extremely difficult) to 10 (extremely easy), while the other answer categories ranged from 1 (agree strongly) to 5 (disagree strongly). The missing data for these indicators did not reach 2%. Statements a, b, and d primarily tap into self-efficacy, item c primarily taps into optimism, and statements e and f primarily tap into resilience [26, 27]. Although in the original concept of PsyCap the dimension of hope is represented; in the ESS6, no direct measure for hope was included [26]. Hope consists of three components: agency, pathways, and goal [28]. Whereas pathways and goals are not well represented in our proxy measure, agency is to some extent reflected in the item "How difficult or easy do you find it to deal with important problems that come up in your life". In addition, optimism–included in the ESS6 –refers partly to hopefulness (namely the emotional facet) [27].

All questions were rescaled to the same range and recoded so that a high score reflects high PsyCap. The indicators related to each subscale of PsyCap were added up to create separate scales for efficacy, resilience, and optimism. We conducted a principal axis factor analysis with varimax rotation of the three subscales. The purpose of the analysis was to test the number of factors needed to adequately describe the data. In the factor analysis we found reasons for constructing a single indicator for PsyCap. The factor extracted had an eigenvalue of 1.79 and accounted for 59.7% of the variance. It was the only factor with an eigenvalue above 1. The factor analysis shows strong convergence of the subscales. The factor loadings on the first factor are .78 for efficacy, .61 for optimism and .50 for resilience. In addition, the Cronbach's alpha for the PsyCap scale was close to .70 (Cronbach's $\alpha$ = .64). When calculating the Cronbach's $\alpha$ for each country separately, these were all close to or higher than .70 (except for Switzerland (.34), Italy (.52), Portugal (.53), Russia (.59) and Kosovo (.55)). Yet, when doing the factor analysis for each country separately, each of the 29 countries showed only one factor with an eigenvalue above 1. Based on these empirical considerations, we are confident that the subscales point to a unified underlying construct. Therefore, three subscales were summed to

create the PsyCap scale (range 0 to 1, a high value represents a high PsyCap). For the LCA, this variable was recoded into three categories based on its tertiles: 'low'; 'middle' and 'high'.

**Bourdieusian capitals.** Previous research [13, 17, 22] offered the groundwork for the selected measurement items used in our study. To adequately test the transferability between different capitals, we selected measures that distinctly represent only one form of capital. Therefore, items that reflect two capital dimensions are excluded. For instance, occupational status reflects both cultural capital (as educational attainment often is a prerequisite for access to certain professions) and economic capital (as a job is a primary means of income and financial security) and is subsequently omitted from our analyses. In addition, we limited the number of measurement items for each capital dimension to maximum two to balance the LCA and simplify its interpretation, while still capturing the essence of each capital type.

**Cultural capital** was operationalized using respondents' *educational level (missing values .6%)*. The respondents were grouped into three educational categories according to the International Standard Classification of Education (ISCED): 'low' (up to lower secondary); 'medium' (up to post-secondary non-tertiary); and 'high' (completed tertiary education). We also included the *parent's education (missing values 5.5%)*, which is the level of educational attainment of the respondent's most highly educated parent (see also [17]), with three categories (based on ISCED); 'low' (up to lower secondary); 'medium' (up to post-secondary non-tertiary); and 'high' (completed tertiary education). We include the educational level of the most highly educated parent, because cultural capital is to a large extent embodied within the parent and through upbringing transferred [29].

**Economic capital** is measured using the respondents' financial situation. This is measured using a question on the *perception of the current household income* being sufficient or not (*missing values 1.5%*). Measuring income sufficiency at the household level is appropriate, since income–although related to individual's situations–is a concern mostly situated at the household level. A partner's income might make up for a low income, or a high income might be insufficient because of high costs. The answer categories were: (1) Living comfortably on present income; (2) Coping on present income; (3) Difficult on present income and (4) Very difficult on present income. This variable was reversed so that a high value represents the best financial situation and categories 3 and 4 were taken together to simplify the interpretation of the models. Income measured at the household level has been done previously in social class analyses [17]. In our analyses, the net income of the household was not included due to many missing values (over 20%).

**Social capital** was operationalized using two indicators, weak and strong ties [30]. Firstly, a question measures *strong ties (*missing values .9%*)*: "To what extent do you receive help and support from people you are close to when you need it?" Answer categories range from 0 (not at all) to 6 (completely). For the LCA, this variable was recoded into three categories based on its tertiles: 'low' (0 to 4); 'middle' (5) and 'high' (6). Secondly, a question measures *weak ties (missing values 3.4%)*: "To what extent do you take part in social activities compared to others of same age?" The answer categories were: (1) much less than most; (2) less than most; (3) about the same; (4) more than most and (5) much more than most. For the LCA, this variable was recoded into three categories: (much) less than most; about the same and (much) more than most.

## Analyses

Our total sample comprised 35,313 respondents (non-weighted, with listwise deletion of missing data). We first described the population using percentages, means and standard deviations (SD). These analyses were performed using SPSS version 28.

To test whether PsyCap is a form of capital (Objective 1), some conditions needed to be fulfilled, which were: (1) it needed to be correlated to the other types of capital. By the same token (2) the correlation should not have been too high, as this would have made the idea of adding another type of capital redundant. We deployed different modes of analysis to investigate this. First, we tested Spearman rank correlations between PsyCap and the indicators that make up the Bourdieusian capitals. To account for the nested structure of our cross-national data, we computed multilevel Spearman correlation coefficients. We deemed correlation coefficients that were between zero and $< .2$ as weak, between .2 and $< .4$ as moderate and above that as strong [31]. These analyses were performed with the correlation package [32], version 0.6–12 using the R software [33].

Second, using Confirmatory Factor Analysis (CFA), we tested whether the scales of PsyCap were empirically distinct from the indicators belonging to the other forms of capital. To that end, we compared the fit of four competing models. In the first model, we let all items of PsyCap load onto the latent variable for cultural capital, while also including the single-item for economic capital and the latent variable for social capital. In the two subsequent models, the items of PsyCap loaded on a latent variable including the single item of economic capital on the one hand, and on the latent variable for social capital on the other hand (while also including the other types of capital in the model). In the last model, we let the items of PsyCap load onto the latent variable PsyCap, while also including a single item for economic capital and two latent variables for cultural capital and social capital. We assessed the fit of the different models using the following fit indices: root mean square error of approximation (RMSEA) $\leq .05$ indicated a good fit to data and $.05 < \text{RMSEA} < .08$ indicated a satisfactory fit [34]. Comparative fit index (CFI) $\geq .95$ and standardized root mean square residual (SRMR) $< .09$ indicated a good fit to data [34]. This was investigated with the Lavaan package [35], version 0.6–12 using the R software [33].

The different types of social positions based on the four distinct capitals (Objective 2) were assessed using LCA [36] (Objective 2). For this, Latent Gold 5.1[TM] software was used. LCA [37] was conducted on the six capital indicators. Moreover, we have taken the nested structure of our data into account, by including direct effects of the country variable on item responses. As a result, the unique meaning of the item responses is filtered out by the direct effects of the country indicator on each item [38].

LCA uses the distribution of the indicators over the sample to create an empirical typology of–in this case–social positions among the European population. In other words, individuals included in the sample are rearranged in a limited number of groups (classes), based on their degree of similarity on manifest indicators reflecting their capital situation. There are social positions in the population, but they are unobserved (latent). We start with a one-class model and then fit $n$ models. For selecting the final model and subsequently the number of social classes, two criteria are taken into consideration: (1) the best-fitting model based on statistical criteria and (2) theoretical meaning of the model [39]. Regarding the former, the best-fitting model is obtained by evaluating the Akaike Information Criterion (AIC), the Bayesian Information Criterion (BIC) and the Consistent Akaike Information Criterion (CIAC) [37]. The lower the AIC, BIC, and CAIC, the better the model fits the data (that is, the more accurate the relationships predicted by the model represent the real pattern of relations observed in the data). When comparing the changes in AIC, BIC, and CAIC for each model, a clear drop in improvement of the changes suggests model saturation [39]. Regarding the theoretical interpretation, we examine the relations between the latent classes and capital-indicators (the conditional probabilities) for three statistically acceptable models to assess the meaningfulness of each extra latent class.

Finally, the country-specific distribution of the social class typology is described (Objective 3), using cross-tabulations of the mean cluster probabilities. To test our hypotheses, we have included the Gross Domestic Product (GDP) per capita (in US $), the GINI index and the total

government expenditure on education (% of GDP) for each country in 2012, to represent a country's economic development, income inequality and educational expansion, respectively. These indicators were obtained from the World Bank [40].

Throughout all analyses, data have been weighted using the analysis weights provided by the ESS6, resulting in a corrected sample size of 35,489. These weights correct for differential selection probabilities within each country as specified by sample design, for nonresponse, for noncoverage, and for sampling error related to the four post-stratification variables and considers differences in population size across countries [41].

## Results

### Descriptive analyses

Table 1 shows the general characteristics of the sample. There are 48.1% men and 51.9% women in the sample and the mean age is 43.2 years ($SD$ = 12.8). The mean of PsyCap is 0.65

**Table 1. Frequency table, ESS6, 29 countries, age 21–65, weighted (N = 35,489).**

| Indicator | N | % |
|---|---|---|
| **Gender** | | |
| Male | 17,067 | 48.1 |
| Female | 18,422 | 51.9 |
| **Age** | 43.2 (mean) | 12.8 ($SD$) |
| **PsyCap (range 0–1)** | .65 (mean) | .15 ($SD$) |
| Low tertile | 11,754 | 33.1 |
| Middle tertile | 12,152 | 34.2 |
| High tertile | 11,583 | 32.6 |
| **Cultural capital** | | |
| **Own education** | | |
| Low | 12,859 | 36.2 |
| Middle | 14,692 | 41.4 |
| High | 7,939 | 22.4 |
| **Parent's education** | | |
| Low | 15,771 | 44.4 |
| Middle | 14,471 | 40.8 |
| High | 5,247 | 14.8 |
| **Economic capital** | | |
| **Perception of income** | | |
| (Very) difficult | 12,028 | 33.9 |
| Coping | 15,761 | 44.4 |
| Comfortable | 7,700 | 21.7 |
| **Social capital** | | |
| **strong ties (range 0–6)** | 4.92 (mean) | 1.25 ($SD$) |
| Low tertile | 9,716 | 27.4 |
| Middle tertile | 11,320 | 31.9 |
| High tertile | 14,453 | 40.7 |
| **weak ties (range 1–5)** | 2.69 (mean) | .93 ($SD$) |
| (much) less than most | 13,301 | 37.5 |
| About the same | 16,712 | 47.1 |
| (much) more than most | 5,476 | 15.4 |

Abbreviation: SD: standard deviation

(*SD* = 0.15). As regards the Bourdieusian capitals, most of the sample is middle educated (41.4%) and their highest educated parent is in most cases low educated (44.4%). Most respondents are coping on the present income (44.4%) and have a good network of strong ties (mean 4.92, *SD* 1.25). Few respondents have a good network of weak ties (15.4% belongs to the group indicating to participate (much) more than most in social activities).

## How does PsyCap relate to other forms of capital?

Table 2 shows the multilevel Spearman correlations. We found weak positive correlations between PsyCap and the indicators for cultural capital (r ≤ .14). Positive moderate correlations were found between PsyCap and the indicator for economic capital on the one hand, and two indicators for social capital on the other hand (r ≤ .24). This suggests that PsyCap can be considered a relatively independent form of capital. Yet, because of the (weak/moderate) overlap, we can also infer that PsyCap is also somewhat related to the same social mechanisms (e.g., transformation) as the Bourdieusian forms of capital.

Table 3 shows that the model with the items of PsyCap loading on a separate latent variable for PsyCap had a significant better model fit than the three capital models. The three-capital model with the PsyCap items loading on a factor "cultural capital" had an inadequate model fit, with the RMSEA-, and CFI-values showing unacceptable levels (See Fig A in S1 Text for the structural model). The three-capital models, with the PsyCap items loading on a factor "economic capital" and "social capital" respectively, had an inadequate model fit with CFI-values < .95 (see Figs B and C in S1 Text for the structural model, respectively). In contrast, the four-capital model with a separate factor for PsyCap shows acceptable levels on the RMSEA-, SRMR-, and CFI-values (see Fig D in S1 Text for the structural model). Based on this information, we conclude that the four-capital model fits our data best.

## Types of social positions based on a four-capital model

Next, we look for different types of social positions based on the four distinct capitals using LCA. Therefore, we firstly need to determine how many types of social positions are present among our respondents. Table 4 provides summary statistics for the LCA models. The table describes models with one to eight latent classes, using the six indicators of capital discussed above. Going up to a 7-class model showed a decrease in change in BIC, AIC, and CAIC. This

**Table 2. Multilevel spearman correlation coefficients between the different indicators of Bourdieusian capital and psychological capital, ESS6, age 21–65, 29 countries, weighted (N = 35,489).**

| | PsyCap (0 → 1) | | Cultural capital | | | | Economic capital | |
| --- | --- | --- | --- | --- | --- | --- | --- | --- |
| | | | Own's education | | Parent's education | | Perception of income | |
| | r | p-value | r | p-value | r | p-value | r | p-value |
| **Cultural capital** | | | | | | | | |
| Own education | .14 | < .001 | *na* | *na* | *na* | *na* | *na* | *na* |
| Parent's education | .11 | < .001 | *na* | *na* | *na* | *na* | *na* | *na* |
| **Economic capital** | | | | | | | | |
| Perception of income | .23 | < .001 | .26 | < .001 | .15 | < .001 | *na* | *na* |
| **Social capital** | | | | | | | | |
| Strong ties | .24 | < .001 | .06 | < .001 | .06 | < .001 | .11 | < .001 |
| Weak ties | .21 | < .001 | .12 | < .001 | .08 | < .001 | .13 | < .001 |

Abbreviations: na: not applicable

**Table 3. Test of model fit of three-factor and four-factor capital models, ESS6, age 21–65, 29 countries, weighted (N = 35,489).**

| | Model fit | | | Change in model fit (4-capital model vs. 3-capital model) | |
|---|---|---|---|---|---|
| | SRMR | RMSEA | CFI | Δχ2/Δdf | p-value |
| Model with items of PsyCap loaded on a factor 'cultural capital' | .07 | .11 | .72 | 2290/3 | < .001 |
| Model with items of PsyCap loaded on a factor 'economic capital' | .04 | .06 | .92 | 260/2 | < .001 |
| Model with items of PsyCap loaded on a factor 'social capital' | .03 | .05 | .94 | 84/3 | < .001 |
| Model with separate factor for PsyCap (4-capital model) | .03 | .05 | .95 | na | na |

Abbreviations: Δχ2: change in Chi-square; Δdf: change in degrees of freedom; na: not applicable

indicates that the 7-class solution gives us an optimal level of model fit/ model parsimony balance. In a next step, we interpreted a 5-, 6- and 7-class solution. We were able to make the most sensible interpretation using the 6-class solution. When comparing the 5-class to the 6-class solution, the 6-class solution adds a profoundly different profile. Yet, when comparing the 6-class to the 7-class solution, the 7-class solution adds a profile with a very small cluster size (6%) and no profoundly different profile. Based on this information, we decided that the 6-class model best represents reality.

Table 5 shows the conditional probabilities for the final capital typology, i.e., the 6-class solution. These conditional probabilities contain information on the associations between the latent capital types and the constituting manifest proxy-indicators. Based on these relationships, the types of social positions are given their substantive interpretation, which is reflected in their names.

The first capital-constellation, *the high PsyCap/social/economic capital group*, stands out due to their high score on PsyCap and relatively high probabilities of receiving help from close contacts and engaging in (many) more social activities. This class exhibits the highest probabilities of coping on the present income and higher probabilities than the overall mean of living comfortable on the present income. They show low probabilities of belonging to the high-educated and high-educated parents' group. This group compromises 29% of the sample.

The second capital-constellation type, labelled as the *low-educated deprived*, exhibits generally low levels of capital, for all types of capital. This class comprises 25% of the sample. Their probabilities of belonging to the low-educated and the low-educated parents' group are higher, compared to other classes. They have the highest probability of finding it (very) difficult to cope with the present income, compared to other clusters. Their social capital is marked by the highest probabilities of belonging to the tertile with the least experiences of receiving help and

**Table 4. Comparison of selected fit indices and degree of model improvement over the different latent class models, ESS6, age 21–65, 29 countries, weighted (N = 35,489).**

| Model | BIC | AIC | CAIC | ΔBIC | ΔAIC | Δ CAIC |
|---|---|---|---|---|---|---|
| 1 class | 432857.6 | 431331.8 | 433037.6 | | | |
| 2 classes | 418877.4 | 417054.9 | 419092.4 | 13980.2 | 14276.9 | 13945.2 |
| 3 classes | 416234.9 | 414115.6 | 416484.9 | 2642.6 | 2939.3 | 2607.6 |
| 4 classes | 414817.8 | 412401.8 | 415102.8 | 1417.1 | 1713.8 | 1382.1 |
| 5 classes | 413785.5 | 411072.8 | 414105.5 | 1032.3 | 1329.0 | 997.3 |
| 6 classes | 413312.9 | 410303.5 | 413667.9 | 472.6 | 769.3 | 437.6 |
| 7 classes | 413054.1 | 409748.1 | 413444.1 | 258.7 | 555.4 | 223.7 |
| 8 classes | 413180.5 | 409577.7 | 413605.5 | -126.3 | 170.4 | -161.3 |

**Table 5. 6-class model: Distribution of class conditional probabilities over capital indicators, 29 countries, age 21–65, weighted, ESS6 (N = 35,489).**

| | Class 1 | Class 2 | Class 3 | Class 4 | Class 5 | Class 6 | Overall |
|---|---|---|---|---|---|---|---|
| | *high PsyCap/social/ economic capital* | *Low-educated deprived* | *Middle-educated deprived* | *Upward well-off* | *Generationally well-off* | *high PsyCap/social/ cultural capital* | |
| **Cluster Size** | 29% | 25% | 16% | 11% | 10% | 9% | |
| **Indicators** | | | | | | | |
| **PsyCap** | | | | | | | |
| Low tertile | 0.1441 | 0.5618 | 0.5764 | 0.2348 | 0.2621 | 0.0808 | 0.3312 |
| Middle tertile | 0.3497 | 0.3255 | 0.3198 | 0.3884 | 0.3915 | 0.2953 | 0.3424 |
| High tertile | 0.5062 | 0.1126 | 0.1038 | 0.3768 | 0.3465 | 0.6239 | 0.3264 |
| Mean | 2.3622 | 1.5508 | 1.5275 | 2.142 | 2.0844 | 2.5430 | 1.9952 |
| **Cultural capital** | | | | | | | |
| **Own's education** | | | | | | | |
| Low | 0.5294 | 0.6986 | 0.1763 | 0.0006 | 0.0001 | 0.0809 | 0.3623 |
| Middle | 0.4568 | 0.2969 | 0.7245 | 0.187 | 0.0778 | 0.7027 | 0.4140 |
| High | 0.0139 | 0.0045 | 0.0992 | 0.8124 | 0.9221 | 0.2164 | 0.2237 |
| Mean | 1.4845 | 1.3059 | 1.9230 | 2.8118 | 2.9220 | 2.1355 | 1.8614 |
| **Parents education** | | | | | | | |
| Low | 0.6127 | 0.8329 | 0.0272 | 0.4756 | 0.0005 | 0.0096 | 0.4444 |
| Middle | 0.3799 | 0.1660 | 0.7697 | 0.5072 | 0.2009 | 0.6280 | 0.4078 |
| High | 0.0074 | 0.0011 | 0.2031 | 0.0172 | 0.7985 | 0.3624 | 0.1478 |
| Mean | 1.3947 | 1.1682 | 2.1759 | 1.5416 | 2.7980 | 2.3528 | 1.7034 |
| **Economic capital** | | | | | | | |
| (Very) difficult | 0.2068 | 0.5374 | 0.5633 | 0.1037 | 0.1608 | 0.3200 | 0.3389 |
| Coping | 0.5035 | 0.3965 | 0.3682 | 0.4644 | 0.4549 | 0.4767 | 0.4441 |
| Comfortable | 0.2897 | 0.0661 | 0.0684 | 0.4319 | 0.3843 | 0.2033 | 0.217 |
| Mean | 2.0829 | 1.5288 | 1.5051 | 2.3282 | 2.2235 | 1.8833 | 1.8781 |
| **Social capital** | | | | | | | |
| **Strong ties** | | | | | | | |
| Low tertile | 0.1432 | 0.4047 | 0.4615 | 0.2130 | 0.2471 | 0.1208 | 0.2738 |
| Middle tertile | 0.2982 | 0.3391 | 0.3270 | 0.3326 | 0.3367 | 0.2817 | 0.319 |
| High tertile | 0.5586 | 0.2562 | 0.2115 | 0.4544 | 0.4162 | 0.5975 | 0.4072 |
| Mean | 2.4154 | 1.8515 | 1.7500 | 2.2414 | 2.1691 | 2.4767 | 2.1335 |
| **Weak ties** | | | | | | | |
| (much) less than most | 0.2780 | 0.5437 | 0.4357 | 0.3152 | 0.3309 | 0.2428 | 0.3748 |
| About the same | 0.5098 | 0.392 | 0.4571 | 0.5037 | 0.4991 | 0.5137 | 0.4709 |
| (Much) more than most | 0.2122 | 0.0643 | 0.1071 | 0.1811 | 0.1700 | 0.2435 | 0.1543 |
| Mean | 1.9341 | 1.5206 | 1.6714 | 1.8659 | 1.8391 | 2.0007 | 1.7795 |
| **Capital volume*** | 11.6738 | 8.9258 | 10.5529 | 12.9309 | 14.0361 | 13.392 | 11.3511 |

* Capital volume = sum of means of the six capital-indicators

engaging in (much) less social activities than most. In addition, their mean score on PsyCap is notably lower compared to the overall mean.

The third capital-constellation, the ***middle-educated deprived***, is characterized by the lowest mean score on PsyCap compared to the overall mean. This class shows high probabilities of having middle-educated parents and being middle-educated themselves. This class also demonstrates the highest probabilities of finding it (very) difficult to cope with the present income

and the lowest probabilities of belonging to the tertile with the best experiences of receiving help, compared to the overall mean. Finally, they exhibit the second-highest probabilities of engaging in (far) fewer social activities than most. Approximately 16% of the sample falls into this class.

The fourth capital-constellation, referred to as the ***upward well-off***, is characterized by the highest probabilities of tertiary education, alongside the lowest probabilities of having high-educated parents. They exhibit the highest probabilities of living comfortably on the present income. In terms of strong ties, they have slightly higher probabilities of belonging to the middle and high tertile group, compared to the overall mean. Regarding weak ties, their probabilities of engaging in about the same or (many) more social activities than most are slightly higher, compared to the overall mean. Additionally, their mean score on PsyCap is the third highest among all clusters. This group consists of approximately 11% of the sample.

The fifth capital-constellation, labeled the ***intergenerationally well-off***, is distinguished by the highest probabilities of tertiary education and having tertiary-educated parents. They also display the second highest probabilities of living comfortably on the present income. In terms of strong ties, they show slightly higher probabilities of belonging to the middle and high tertile group, compared to the overall mean. As regards weak ties, they exhibit slightly higher probabilities of engaging in about the same or (many) more social activities than most, compared to the overall mean. Their mean score on PsyCap is slightly higher than the overall mean. This group consists of approximately 10% of the sample.

Finally, ***the high PsyCap/social/cultural capital group***, is marked by their highest mean score on PsyCap and the highest probabilities of positive experiences of receiving help and engaging in (much) more social activities. Otherwise, this class seems to show low probabilities of living comfortably on the present income. They also exhibit low probabilities of belonging to the low-educated and low-educated parents' group. This group consists of approximately 9% of the sample.

### Country-specific distribution

Table 6 shows clear differences between the 29 European countries regarding the distribution of class probabilities. Within economically less developed countries, i.e., with low GDP per capita (such as the Eastern and South-Eastern European countries), the probability to belong to the high PsyCap/social/economic capital class is zero or close to zero and the probability to belong to Middle-educated deprived class is high. In economically developed countries, the probability to belong to the high PsyCap/social/economic capital class and the two well-off social classes is higher, compared to their counterparts. However, even though the probability to belong to the Upward well-off class is the highest in Norway, it is also high in Portugal, Spain, and Poland. Moreover, a high probability to belong to the Generationally well-off class is also found in Hungary, Slovenia, and Slovakia. For countries with a (relatively) low GDP per capita and low expenditure on education (such as Southern and South-Eastern European countries) and for economically developed countries with high income inequality, the probability to belong to the Low-educated deprived class is high. Finally, the highest probabilities of belonging to the high PsyCap/social/cultural capital class are found in Finland and Eastern European countries.

### Discussion

The aim of this paper was to assess whether PsyCap can be empirically integrated into the Bourdieusian capital framework and to investigate the existing social positions using this extended capital approach in a large and representative European database. Regarding the first

**Table 6. Distribution of the class probabilities over four-capital social classes within 29 countries for persons between 21 and 65 years old, ESS6.**

| | Macro-indicators | | | Class probabilities | | | | | |
|---|---|---|---|---|---|---|---|---|---|
| | GDP per capita (US$) | GINI | Educational Expansion[1] | Low-educated deprived | Middle-educated deprived | High Psy/ soc/eco cap. | Upward well-off | High Psy/ soc/cult cap. | Generationally well-off |
| Kosovo | 3,410 | 29.0 | n.a. | 47 | 8 | 15 | 4 | 25 | 2 |
| Ukraine | 4,004 | 24.7 | 6.44 | 10 | 37 | 8 | 0 | 35 | 10 |
| Albania | 4,247 | 29.0 | 3.31 | 70 | 1 | 6 | 15 | 3 | 4 |
| Bulgaria | 7,430 | 36 | 3.48 | 35 | 20 | 0 | 2 | 31 | 12 |
| Hungary | 12,984 | 30.8 | 4.14 | 24 | 50 | 0 | 0 | 6 | 20 |
| Poland | 13,010 | 33.0 | 4.86 | 35 | 1 | 30 | 24 | 1 | 8 |
| Lithuania | 14,367 | 35.1 | 4.76 | 33 | 23 | 0 | 4 | 31 | 9 |
| RussianFederation | 15,420 | 40.7 | 3.79 | 23 | 38 | 12 | 0 | 23 | 3 |
| Estonia | 17,403 | 32.9 | 4.72 | 17 | 22 | 2 | 5 | 39 | 16 |
| Slovakia | 17,498 | 26.1 | 3.86 | 15 | 57 | 0 | 0 | 9 | 18 |
| Czechia | 19,870 | 26.1 | 4.22 | 1 | 62 | 0 | 0 | 24 | 13 |
| Portugal | 20,563 | 36.0 | 4.95 | 54 | 0 | 7 | 33 | 0 | 7 |
| Slovenia | 22,641 | 25.6 | 5.62 | 17 | 36 | 4 | 0 | 25 | 18 |
| Spain | 28,322 | 35.4 | 4.47 | 36 | 1 | 22 | 26 | 0 | 15 |
| Cyprus | 28,910 | 34.3 | 5.92 | 42 | 11 | 18 | 22 | 0 | 7 |
| Israel | 33,156 | 41.3 | 5.59 | 29 | 14 | 26 | 11 | 9 | 11 |
| Italy | 35,051 | 35.2 | 4.06 | 44 | 10 | 27 | 7 | 5 | 8 |
| France | 40,870 | 33.1 | 5.46 | 23 | 4 | 51 | 12 | 2 | 9 |
| United Kingdom | 42,497 | 33.1 | 5.63 | 22 | 6 | 38 | 20 | 1 | 13 |
| Germany | 43,855 | 31.1 | 4.93 | 16 | 0 | 58 | 18 | 2 | 7 |
| Belgium | 44,670 | 27.5 | 6.26 | 25 | 13 | 26 | 17 | 1 | 19 |
| Iceland | 45,995 | 26.8 | 7.58 | 0 | 17 | 39 | 24 | 6 | 13 |
| Finland | 47,708 | 27.1 | 7.15 | 0 | 5 | 39 | 6 | 41 | 9 |
| Ireland | 48,943 | 33.2 | 6.16[2] | 32 | 7 | 37 | 15 | 0 | 8 |
| Netherlands | 50,070 | 27.6 | 5.41 | 36 | 7 | 22 | 20 | 0 | 15 |
| Sweden | 58,037 | 27.6 | 7.57 | 11 | 15 | 39 | 15 | 7 | 12 |
| Denmark | 58,507 | 27.8 | 7.24 | 6 | 2 | 48 | 17 | 10 | 15 |
| Switzerland | 85,836 | 31.6 | 4.90 | 10 | 0 | 59 | 20 | 0 | 10 |
| Norway | 102,175 | 25.7 | 7.33 | 0 | 10 | 33 | 33 | 6 | 18 |

[1]Government expenditure on education, total (% of GDP); [2] Data from 2009 (data 2012 unavailable); n.a.: not available

aim, our study provides empirical support to recent theoretical claims calling to include Psy-Cap into Bourdieu's capital framework [3, 4]. To the best of our knowledge, no scholars have empirically tested these theoretical claims yet.

The integration of PsyCap within Bourdieu's framework contributes to addressing social inequalities in various ways, both from a psychological and a sociological perspective. Regarding the former, our results emphasize that understanding differences in PsyCap must go beyond attributing them to individual differences and individual efforts, and needs to incorporate external, societal influences. We have highlighted between-group differences in PsyCap where some groups consistently exhibit higher average levels, probably due to shared, beneficial social and contextual factors. Ignoring structural, between-group differences in PsyCap downplays the impact of enduring social inequalities on people's psychological resources and well-being [4]. Thus, for psychological research, our results call for a shift in research focus

from the intra- and inter-individual level to understanding PsyCap as a social and group-level phenomenon as well.

From a sociological perspective, our study offers two key contributions for research on social inequalities. Firstly, our study advocates the use of positive psychological states as explanatory variables rather than outcomes, a departure from the common practice in the current state of sociological research. This perspective views positive psychological states as a resource and a principle of distinction, unequally distributed in society, with potential investments for returns [4]. Secondly, integrating PsyCap into the Bourdieusian framework can help to address the longstanding issue of understanding the relationship between social and individual differences in the study of social inequalities. Sociological studies often focus on the intergenerational transmission and reproduction of social differences, but the concept of PsyCap introduces the idea of individual differences (as it has a trait-like component), providing a better framework to understand variations within families [4]. More specifically, it may help to explain intra-familial differences in social mobility patterns.

Finally, our study adds to the ongoing discussion in psychological science on how to appropriately conceive and measure social class. At the moment, this matter remains unsettled, with psychologists rarely defining social class theoretically [42]. We offer a theoretical underpinning of social class combining insights from both classical sociological theory and psychology. Beneficial about the capital approach is that it focusses more on the underlying dimensions than the exact combinations (i.e., class fractions) present in a specific population. This avoids unproductive discussions about whether specific classifications are still relevant and makes it easier to study processes of class formation with data from different contexts. Indeed, one should not forget that a key characteristic of the capital approach is that it draws attention to the underlying transformation *processes* rather than the specific outcomes. This reorientates the question from *why is there social inequality* to *why is social inequality so persistent*.

In addition, within Europe, we found six types of social positions (Objective 2): Two types with overall high capital volume and two with overall low capital volume. Two social positions have an "imbalanced" capital composition: With relatively high levels of PsyCap and social capital, but varying levels of economic and cultural capital. The existence of the latter social positions has several implications. They are, for example, clearly relevant in light of the insights derived from resistance theories (see amongst others Knight Abowitz (2000)) for defining social classes. According to these theories, social groups in a weak social position rely on different coping strategies to make life bearable. One of these coping mechanisms is the participation in a subculture or an in-group community in which the (otherwise vulnerable) self-image of its members can be protected [43]. Our findings suggests that such coping and compensation strategies might be at work for those deprived of cultural or economic capital and that these strategies might provide groups of a relatively high volume of capital (or means of power) in society. This finding also underscores the importance of the association between social capital and PsyCap. Social interactions heavily influence people's self-image, sense of self-efficacy, and their general view on the future [44]. Likewise, people with a high volume of PsyCap are more likely to initiate and develop resourceful social relations with others [6].

As regards the deprived types, we found a low-educated and a middle-educated class. The overall low volumes of capital of these social positions can be linked to, amongst others, the centrality of education in what we call the "schooled society" and the importance of education in the access to a wide range of outcomes, including aspects of socio-economic and cultural position [45]. Education can promote feelings of entitlement, that is, the feeling that one deserves to be successful in life, which in turn might lead to positive states like self-confidence etc. [46].

Lastly, two well-off types were found. Because high PsyCap is both the result and the origin of success experiences, it is not surprising that relatively high PsyCap is found among people

in well-off types of social positions. These results highlight that PsyCap is unequally distributed in society and that PsyCap is a form of capital, which helps explain the endurance of social hierarchies.

Finally, our results also point in the direction that country policies and economic conditions play a significant role in the development and distribution of social classes. More than half of the European population has no tertiary education, reflected in large classes with low cultural capital: the high PsyCap/social/economic capital class and the two deprived classes. In countries where governments allocate large funds to education systems (like Norway, Finland and Denmark) [16], classes with low cultural capital were less prevalent. In line with our hypotheses, in economically developed countries like Western and Northern Europe, the high PsyCap/social/economic capital class was common. This suggests that these countries can facilitate a relatively well-off social position, despite low cultural capital. In economically developed countries with high GINI indexes, the prevalence of deprived classes for low-educated is noteworthy. In such contexts, when insufficient initiatives are undertaken to mitigate income inequality, stark distinctions emerge between those possessing cultural capital and those without. Yet, more research is needed to dig deeper into differences in social classes distributions across countries.

Several limitations are present within this study. Firstly, the cross-sectional data of 2012 can be considered relatively old. However, the unique module on personal well-being of the ESS6 is to our knowledge the most suitable available data source for our research objectives. Secondly, using ESS6 we were unable to capture all aspects of the Bourdieusian capitals, such as objectified cultural capital (e.g., the possession of books or paintings), embodied cultural capital (e.g., tastes in music and art), or alternative measurement approaches for economic capital (e.g., the ownership of valuable goods, such as a home or a car) [11]. Regarding social capital, ESS6 does provide more indicators (such as civic participation or trust) [13]. Yet, to keep the models balanced, we decided to utilize a maximum of two indicators for each type of capital. The inclusion of alternative measurement approaches would allow a more nuanced and comprehensive view of the Bourdieusian capitals, which could impact the social class cluster solution, e.g., showing more nuanced within-class differences. Unfortunately, we were unable to assess the potential impact of including alternative measurement approaches.

Additionally, the indicator for PsyCap is only a proxy for the underlying theoretical concept and the measurement of the dimension hope was suboptimal. The lack of 'hope' in our PsyCap construct can potentially skew our results as PsyCap is a second-order construct based on shared commonalities of first-order psychological resources (hope, resilience, efficacy, and optimism), but also based on their unique characteristics [47]. The discriminant validity of these constructs has been empirically established [48]. Unfortunately, the ESS data do not include a proxy for the 'hope' element of PsyCap. Given that we used PsyCap to define clusters, our analysis may have missed a social class cluster distinctly characterized by waypower, which is unique to hope. This 'overlooked' social class might also have a specific cross-country distribution as the level of hope varies between countries and may be related to policies like government spending on education and social services [49]. Whether such a cluster exists and how it is distributed over countries is an empirical question that we cannot answer with the available data. There are, however, certain indications that suggest that the component 'hope' does not distinctly dominate profiles within populations. Indeed, studies based on person-centered latent profile analyses of the PsyCap components show a gradient in the profiles (ranging from low, to moderate to rich (general) PsyCap) [50–52] or in other cases the other PsyCap components (optimism or resilience) seem to dominate the profiles [53, 54]. Our research shows that there is much to be gained by including PsyCap in comparative research and recommends that a more comprehensive measure for PsyCap is included in large-scale survey projects like the ESS.

Thirdly, we were unable to use the objective measure of income, because of a high number of missing values. Yet, objective income is very much influenced by the country one lives/works in. This might be problematic in a transnational study because one would capture national differences rather than differences in economic capital. Fourth, the scope of this article included all individuals aged 21 to 65. Yet, other demographic groups might be of interest, such as specific age groups (e.g., those at the beginning of their career), the retired, workers, children, etc. Finally, there is the issue of how the LCA solution (or social class typology) would be different if tested within each country separately. Unfortunately, doing a cross-national validation fell out of the scope of this research, as the primary focus was to find support for previous theoretical claims and centered on examining the broad applicability of the social class framework. Nevertheless, we allowed for direct country effects on the indicators when estimating the latent structure. Despite these limitations, the use of a large representative European dataset ensures the generalizability of our results to the European population.

## Conclusion

Our study provides empirical support, using a large representative European dataset of 29 countries, to recent theoretical claims calling to include PsyCap into the Bourdieu's capital framework. Six different social fractions were found, based on the 'integrated' social class framework, in Europe. These social fractions show that high PsyCap is unevenly divided among social groups. Our results indicate that country policies and economic conditions significantly influence the development and distribution of social classes.

## Supporting information

**S1 Text. Structural models.**
(DOCX)

## Acknowledgments

The authors would like to thank Julie Vanderleyden for help with the cluster analyses.

## Author Contributions

**Conceptualization:** Deborah De Moortel, Mattias Vos, Bram Spruyt, Christophe Vanroelen, Joeri Hofmans, Edina Dóci.

**Formal analysis:** Deborah De Moortel, Mattias Vos.

**Methodology:** Deborah De Moortel, Christophe Vanroelen, Joeri Hofmans.

**Visualization:** Mattias Vos.

**Writing – original draft:** Deborah De Moortel.

**Writing – review & editing:** Mattias Vos, Bram Spruyt, Christophe Vanroelen, Joeri Hofmans, Edina Dóci.

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
