## [Decision Letter · Decision Letter 0]

7 Nov 2023

PONE-D-23-17518Why can’t we just be more positive? A capital approach towards positive psychological states and their role for explaining social inequalitiesPLOS ONE

Dear Dr. De Moortel,

Thank you for submitting your manuscript to PLOS ONE. After careful consideration, we feel that it has merit but does not fully meet PLOS ONE’s publication criteria as it currently stands. Therefore, we invite you to submit a revised version of the manuscript that addresses the points raised during the review process. As I compile the review comments as the editor, both reviewers suggest complementary works on a methodological front; one reviewer additionally recommends that you enrich the sociological theoretical context and consequence/contribution of the manuscript. I sympathize with them, and ask you for revision.

We look forward to receiving your revised manuscript.

Kind regards,

Jae-Mahn Shim

Academic Editor

PLOS ONE

Journal Requirements:

"This research is facilitated by the research grant ‘FWO1.2.T82.21N’, which is assigned to the first author by the Research Foundation Flanders. The authors would like to thank Julie Vanderleyden for help with the cluster analyses."

"This research is facilitated by the research grant ‘FWO1.2.T82.21N’, which is assigned to the first author by the Research Foundation Flanders. The funders had no role in study design, data collection and analysis, decision to publish, or preparation of the manuscript."

Reviewers' comments:

Reviewer's Responses to Questions

**Comments to the Author**

1. Is the manuscript technically sound, and do the data support the conclusions?

Reviewer #1: Partly

Reviewer #2: Yes

2. Has the statistical analysis been performed appropriately and rigorously? 

Reviewer #1: I Don't Know

Reviewer #2: Yes

3. Have the authors made all data underlying the findings in their manuscript fully available?

Reviewer #1: Yes

Reviewer #2: No

4. Is the manuscript presented in an intelligible fashion and written in standard English?

Reviewer #1: Yes

Reviewer #2: Yes

5. Review Comments to the Author

Reviewer #1: The study investigates whether Psychological Capital can be seen as part of the Bourdieusian capital framework including, social, cultural and economical capital. The researchers used existing European panel data to operational the 4 different capitals and investigated 1) whether a confirmatory factor analysis could show a better fit for a 4 factor structure in the data than for a 3 factor structure (whether PsyCap is related to other capitals but conceptually distinct from them and therefore should be included when 'defining' social class). The authors indeed found evidence for this 4 factor structure. Thereafter, they performed latent profile analyses using the 4 different capitals to see whether the data shows latent subgroups. Results showed 6 different latent subgroups, with a different composition on scores on the 4 different capitals included. the researchers also should descriptive results for the different countries included.

The manuscript is well written and contributes to theory development in relation to social class. A large sample was used including several European countries. However, what diminished my enthusiasm for the paper was the following:

1. In the Introduction section, the authors do not explain why they included several European countries in their study, and also do not discuss that the way social class is defined also depends on context. Later on, also country specific results are discussed, and I think this could be also included in the aim of the study.

2. The Methods and analyses used are, to my knowledge, sophisticated. Nevertheless, the section on how authors included the survey weight is unclear (was this weight determined by the authors or by researchers of the European social survey (if yes, I miss a reference), what exactly is controlled for in this weight?). That 2 times a different N is presented in the first paragraph of the analyses is also unclear (how come the first N is lower then the second one??).

3. I was wondering why the authors did not transform their items into ridit-scores, so they can actually be compared over countries. I think, if I understood it correctly, the authors now base their categorization of scores in to low-mid-high on the different items measuring capital on pooled data from all the European countries together. It would have been more comparable over countries to first calculate ridit scores for each item per country before including them into one analyses. The authors only argue that objective income could be influenced by the country one lives in, so it might even be 'better' to use the subjective measure they used, but this is also true for other capitals and it would have been better to take this into account when performing analyses.

4. A lot of the Discussion section is also description of results and is very long. I think this can be more concise and also for a great part presented under results.

5. I think authors use very firm description of and statements about the different latent clusters (also implying causality) in their result and discussion section, however the different items used to measure the different capitals are not optimal (as mentioned in the discussion as a limitation) and it is cross-sectional/correlational data, making statements about causality impossible. the authors should really be more careful in formulating their conclusion. This will also make the discussion more concise.

Reviewer #2: The paper addresses a significant research gap by proposing the integration of psychological capital (PsyCap) into Pierre Bourdieu's capital framework to better understand social inequality. The paper demonstrates a comprehensive understanding of the theoretical background and offers well-defined research objectives. However, there are two areas that need further attention to enhance the paper's quality before publication in PLOS One.

Measurement Tools for Capital

- The paper raises a valid concern about the limited measurement tools for different capitals (31). As the authors suggested, it is essential to acknowledge the limitations of the selected measurement tools and consider exploring alternative methods to ensure the robustness of the findings.

- The paper would benefit from a discussion of potential alternative measurement approaches to strengthen the validity of the research results. For instance, if objective income measures are not comparable across countries, the authors may consider income percentile calculator depending on each country’s income distribution.

- Similarly, while the authors do acknowledge the limitations of cultural capital measures, I still believe it is crucial to discuss the anticipated outcomes if additional indicators are incorporated.

- Due to the potential for results to vary significantly depending on the chosen measurement tool, I believe it is necessary to validate using a wider range of measurement tools.

Significance and Implications

- While the paper rightly acknowledges the research gap in empirical testing of PsyCap within Bourdieu's framework, it could provide a more explicit discussion of the significance of this integration. How can this integration contribute to addressing social inequality and understanding social classes? Clarifying the broader implications of the research would make the paper more compelling.

- For instance, when discussing the "invisible" social positions, elaborate on what makes them invisible, particularly within the context of existing sociological theories. Explain how the traditional Bourdieusian framework may overlook these positions and why they are significant.

- While the authors mentioned the issue of cross-national validity, it would be beneficial to elaborate on the implications of this potential limitation. Discuss how cross-national variations in social policy, economic conditions, or cultural factors may influence the distribution of PsyCap and social positions.

6. PLOS authors have the option to publish the peer review history of their article (what does this mean?). If published, this will include your full peer review and any attached files.

Reviewer #1: No

Reviewer #2: No

---

## [Author Response · Author response to Decision Letter 0]

22 Dec 2023

Reviewer 1

The study investigates whether Psychological Capital can be seen as part of the Bourdieusian capital framework including, social, cultural and economical capital. The researchers used existing European panel data to operational the 4 different capitals and investigated 1) whether a confirmatory factor analysis could show a better fit for a 4 factor structure in the data than for a 3 factor structure (whether PsyCap is related to other capitals but conceptually distinct from them and therefore should be included when 'defining' social class). The authors indeed found evidence for this 4 factor structure. Thereafter, they performed latent profile analyses using the 4 different capitals to see whether the data shows latent subgroups. Results showed 6 different latent subgroups, with a different composition on scores on the 4 different capitals included. The researchers also showed descriptive results for the different countries included.

The manuscript is well written and contributes to theory development in relation to social class. A large sample was used including several European countries. 

[Authors]: We want to express our sincere appreciation for Reviewer 1’s time and effort in reviewing our paper. Reviewer 1’s valuable insights and feedback has improved the quality of the work, and we are grateful for the thoughtful comments.

1. However, what diminished my enthusiasm for the paper was the following: In the Introduction section, the authors do not explain why they included several European countries in their study, and also do not discuss that the way social class is defined also depends on context. Later on, also country specific results are discussed, and I think this could be also included in the aim of the study.

[Authors]: Thank you for your thoughtful feedback on our manuscript. 

1) After careful consideration of this remark, we decided to better explain the “transnational” approach of our study and add a third aim to our objectives (a country comparison).

The revised manuscript has three aims: (1) examining the relations of PsyCap and the Bourdieusian capitals; (2) the construction of an integrated social class framework and (3) – as suggested by REVIEWER 1 – a country comparison of the integrated social class framework. 

Only aim 1 and 2 use a transnational approach, which is necessary to construct a broadly applicable social class framework. Research comparing social classes across countries or social class research within a specific country, usually adopts an established social class framework that is applicable broadly, such as the Erikson-Goldthorpe-Portocarero (EGP) social class scheme (Haddon 2015), (neo-)Marxist social class schemes (Kong et al. 2017; Prins et al. 2015), Erik Olin Wright’s social class scheme (Espelt et al. 2008), occupational class models (Hoven et al. 2015), etc. In that sense, the types of social classes that can be found within the population are predetermined based on theoretical grounds (e.g., relations to the means of production, skill gradients, economic sectors, …) and assumed to be valid at least over different high-income countries. Similarly, in our study, we looked for a broadly applicable social class framework, but that is simultaneously able to tap in between country differences. 

Moreover, we decided to use a transnational European sample for aim 1 and 2, because these objectives are mostly to support previous theoretical claims: i.e., assessing whether PsyCap can be integrated in the Bourdieusian capital framework and to scrutinize the social classes generated by this four-capital approach. For these objectives a specific country-context is of lesser importance. Including context variation would lead away from the main aim of these objectives, which is finding evidence for previous theoretical claims. 

Yet, we agree with Reviewer 1 that we did not explain why we included several countries in our manuscript (for aim 1 and 2); therefore, we included the following sentence in the introduction of the revised manuscript: “Using a transnational approach helps to develop a broadly applicable social class framework. Research comparing social classes across countries or social class research within a specific country, usually adopts an established social class framework that is applicable broadly, such as the Erikson-Goldthorpe-Portocarero (EGP) social class scheme (Haddon 2015), (neo-)Marxist social class schemes (Kong et al. 2017), etc. In that sense, the types of social classes that can be found within the population are assumed to be valid over different high-income countries.”

2) As regards Reviewer 1’s remark on how the country-specific context would influence how social class is defined. We want to, firstly, highlight again that the methodology employed in our study is grounded in common methods utilized in social class research, even in cross-national research (Pförtner et al. 2015). Research on social class (inequalities) usually takes an existing social class framework, such as the Erikson-Goldthorpe-Portocarero (EGP) social class scheme (Haddon 2015), to investigate their research aims. These studies, like our study, also have the limitation that different countries exhibit different income inequalities and varying levels of educational democratization (Aamodt & Kyvik 2005), which again affect the sizes of the classes. 

Moreover, our theoretical background, Bourdieu’s capital approach and the PsyCap approach used to scrutinize the social classes, are based on general theoretical concepts (Carmo & Nunes 2013; Donaldson et al. 2020; Wernsing 2013). Therefore, we can use these concepts to study social phenomenon in a transnational context.

Yet we now discuss how social class depends on context, we added extra information before introducing our third aim (see page 10, last paragraph): “Finally, the sizes of social classes within European countries are influenced by a country’s level of income inequality, educational democratization, generosity of the welfare regime, etc. [15,22]. For instance, access to good quality mental health services can enhance an individual’s ability to have high PsyCap [23] and generous replacement income schemes might protect the loss of economic (and subsequently, psychological) capital when unemployed, etc. Therefore, our final aim (objective 3) is to examine cross-national differences in social classes distribution.”

Finally, because we included country differences as an aim, we deleted the gender and age differences in our results. This gives us also the advantage that we were able to reduce the word count of the already lengthy manuscript. Moreover, the overall “story” of the manuscript is now more focussed and cohesive.

We accommodate to REVIEWER 1’s remarks by implementing the subsequent adaptations in the revised manuscript:

- In the introduction, we have provided a more comprehensive rationale for the utilization of all European countries in our analyses. See Page 4, last paragraph.

- In the introduction, we added a final aim which includes presenting country-specific distributions of the ‘integrated’ social class scheme. See page 5, first paragraph.

- In the objectives, we added a final aim referring to the country-specific distribution of the ‘integrated’ class scheme. See Page 10, last paragraph. 

- In the discussion, we elaborate on country-specific distributions of the cluster. See page 29 - final paragraph, page 30 – first paragraph).

- In the discussion, we discuss how the country-specific context would influence how social class is defined. See Page 31, first paragraph.

2. The Methods and analyses used are, to my knowledge, sophisticated. Nevertheless, the section on how authors included the survey weight is unclear (was this weight determined by the authors or by researchers of the European social survey (if yes, I miss a reference), what exactly is controlled for in this weight?). That 2 times a different N is presented in the first paragraph of the analyses is also unclear (how come the first N is lower then the second one??).

[Authors]: Thank you for raising our attention to the use of weights in our research. We agree with Reviewer 1 that the section on weights could have been clearer. The survey weights used in the descriptive and advanced statistical analyses are provided by the European Social Survey. This was made more explicit in the methods section and a reference (Kaminska 2020) was added. See page 17, third paragraph.

Drawing on the guide to using weights with ESS data (Kaminska 2020), which states that “it is recommended that by default you should always use “anweight” (analysis weight) as weight in all analyses”, we used anweight in all our analyses. This weight is suitable for all types of analysis, including when you are studying just one country, when you compare across countries, or when you are studying groups of countries (Kaminska 2020). In the new version of the manuscript, we additionally weighted the CFA models and the Spearman rank correlations (which were unweighted in the original version). 

Anweight corrects for differential selection probabilities within each country as specified by sample design, for non-response, for noncoverage, and takes into account differences in population size across countries (Kaminska 2020). Regarding the latter, this weight corrects for the fact that most countries taking part in the ESS have very similar sample sizes, no matter how large or small their population. Without weighting, any figures combining two or more country’s data would be incorrect, over-representing smaller countries at the expense of larger ones. The mean of the anweight, of my final sample, is 1.0050, the minimum is 0 and the maximum is 20.24.

We present two different N’s ((a) 35,313 and (b) 35,489) in the manuscript. The first number (35,313) is our total sample, non-weighted, with listwise deletion of missing data. The latter number (35,489) is the same sample, but weighted. 

As correctly noticed by Reviewer 1, the weighted sample size is higher than the non-weighted sample size. When weighting data, the sample size can appear higher due to the adjustment applied to certain observations (e.g. some observations in Russia have a value around 20). This adjustment accounts for the varying importance or representation of specific groups or instances in the dataset. Essentially, weighting assigns more influence to certain observations, leading to a perceived increase in the effective sample size. It is crucial to understand that the weighted sample size does not represent the actual number of observations but rather the adjusted size considering the assigned weights.

To make the difference between the numbers clearer, in the table we also put weighted in the title with the number 35,489. In section 2.3. Analyses we wrote: “Our total sample comprised 35,313 respondents (non-weighted, with listwise deletion of missing data)”. And we added a reference to the “Guide on using weights in ESS data” in the methods section.

3. I was wondering why the authors did not transform their items into ridit-scores, so they can actually be compared over countries. I think, if I understood it correctly, the authors now base their categorization of scores in to low-mid-high on the different items measuring capital on pooled data from all the European countries together. It would have been more comparable over countries to first calculate ridit scores for each item per country before including them into one analyses. The authors only argue that objective income could be influenced by the country one lives in, so it might even be 'better' to use the subjective measure they used, but this is also true for other capitals and it would have been better to take this into account when performing analyses.

[Authors]: Thank you for this comment. Regarding your question about ridit-scores: The reviewer seems to suggest that we should remove country-level effects from the data before performing the analyses, because this is essentially what ridit-scores do. 

This would be the way to go if we would be interested in within-country differences only, but we think that those between-country differences are important too (note that when using ridit scores there would be no Figure 1 with the suggested approach because the “average” of each country would be identical). In addition, we account for country-level effects in our analyses by (1) computing multilevel correlation coefficients, and (2) modelling the country-level effect in the LCA. So our take on this is that group-mean centering (by using ridit-scores) would make sense if the hypotheses concern within-country differences only. This is not the case here. Moreover, we do control for country-level differences in our analyses, explicitly taking country-level differences into account. For these reasons, we have not followed this suggestion.

4. A lot of the Discussion section is also description of results and is very long. I think this can be more concise and also for a great part presented under results.

[Authors]: Thank you for raising our attention to the descriptive nature of our discussion section. We agree with Reviewer 1. In the revised manuscript, we rewrote the Discussion section, making it more concise (for instance, deleting the descriptive information and the last two paragraphs of the discussion in the original manuscript) and ensuring that relevant content is appropriately presented under the Results section. We appreciate the opportunity to enhance the clarity and organization of the paper. The discussion section now ends with a short conclusion section for the sake of clarity.

5. I think authors use very firm description of and statements about the different latent clusters (also implying causality) in their result and discussion section, however the different items used to measure the different capitals are not optimal (as mentioned in the discussion as a limitation) and it is cross-sectional/correlational data, making statements about causality impossible. the authors should really be more careful in formulating their conclusion. This will also make the discussion more concise.

[Authors]: We appreciate REVIEWER 1’s observation regarding the firm descriptions and implied causality when describing the different latent clusters. Your point about the limitations of the measurement items and the cross-sectional nature of the data is well taken. We agree with REVIEWER 1 that the relations in our study are correlations. In addition, theoretically, it is also logical that all the relationships are bidirectional (for example, economic capital can be transformed into social capital and vice versa). Therefore, in the results sections, we have rewritten the description of the different latent clusters avoiding the use of causal language. We also exercised caution in our discussion: we thoroughly reviewed the manuscript and examined it for the use of causal language.

Reviewer 2

The paper addresses a significant research gap by proposing the integration of psychological capital (PsyCap) into Pierre Bourdieu's capital framework to better understand social inequality. The paper demonstrates a comprehensive understanding of the theoretical background and offers well-defined research objectives. However, there are two areas that need further attention to enhance the paper's quality before publication in PLOS One.

[Authors]: The authors wish to thank Reviewer 2 for the valuable feedback. We appreciate your recognition of the research gap we aimed to address by integrating PsyCap into Bourdieu's capital framework. We committed to addressing the areas that need improvement (see below).

1) Measurement Tools for Capital: The paper raises a valid concern about the limited measurement tools for different capitals (31). As the authors suggested, it is essential to acknowledge the limitations of the selected measurement tools and consider exploring alternative methods to ensure the robustness of the findings. The paper would benefit from a discussion of potential alternative measurement approaches to strengthen the validity of the research results. For instance, if objective income measures are not comparable across countries, the authors may consider income percentile calcula

---

## [Decision Letter · Decision Letter 1]

22 Mar 2024

PONE-D-23-17518R1Psychological capital and social class: A capital approach to understanding positive psychological states and their role in explaining social inequalitiesPLOS ONE

Dear Dr. De Moortel,

Thank you for submitting your manuscript to PLOS ONE. After careful consideration, we feel that it has merit but does not fully meet PLOS ONE’s publication criteria as it currently stands. Therefore, we invite you to submit a revised version of the manuscript that addresses the points raised during the review process.

We look forward to receiving your revised manuscript.

Kind regards,

Fraide Agustin Ganotice, PhD

Academic Editor

PLOS ONE

Journal Requirements:

Additional Editor Comments:

Dear Authors,

Thank you for submitting your paper titled “Psychological capital and social class: A capital approach to understanding positive psychological states and their role in explaining social inequalities ” to PLOS ONE. Two reviewers have examined the manuscript for which one recommended major revision and one recommended minor revision. I also went over the manuscript and agreed with the two reviewers to recommend minor revision.

Thanks so much.

Respectfully yours,

Fred Ganotice

Reviewers' comments:

Reviewer's Responses to Questions

**Comments to the Author**

1. If the authors have adequately addressed your comments raised in a previous round of review and you feel that this manuscript is now acceptable for publication, you may indicate that here to bypass the “Comments to the Author” section, enter your conflict of interest statement in the “Confidential to Editor” section, and submit your "Accept" recommendation.

Reviewer #1: (No Response)

Reviewer #2: All comments have been addressed

2. Is the manuscript technically sound, and do the data support the conclusions?

Reviewer #1: Yes

Reviewer #2: Partly

3. Has the statistical analysis been performed appropriately and rigorously? 

Reviewer #1: I Don't Know

Reviewer #2: No

4. Have the authors made all data underlying the findings in their manuscript fully available?

Reviewer #1: Yes

Reviewer #2: (No Response)

5. Is the manuscript presented in an intelligible fashion and written in standard English?

Reviewer #1: Yes

Reviewer #2: Yes

6. Review Comments to the Author

Reviewer #1: I have read the revised version of the manuscript with great pleasure. The manuscript has certainly improved, the Results and Discussion section are more concise and have a better focus, and the authors have addressed most of my previous comments in a good way.

Nevertheless, there are a few minor comments still remaining:

The authors response to comment 3 is not completely satisfying. Ridit scores do not remove country-level effects, but make the measures more comparable over countries. For instance, what is considered high wealth or highly educated can differ between countries. When calculating ridit scores for each measure and per country separately, before placing individuals into a low, middle, or high group, makes measures more comparable over countries. In this case you are not comparing oranges with apples. Calculating ridit scores is therefore especially important when using more objective measures. On the other hand, I do not think the results will change much if the authors would have included ridit scores, since they mostly recoded their measures into quite crude groups (categorized indicators into 3 groups (low-mean-high)) and mostly also did not use objective measures but more subjective measures, where one could expect that these should be comparable over countries. However, it might be good to have a statistician have a look, since the sophisticated methods the authors used go beyond my own knowledge.

Regarding the third aim of the paper, where the authors included the suggested aim of examining country differences in the distribution of social classes, it would have been nice if the authors:

a. included hypotheses into the objectives section of the introduction, about the specific country-level indicators they are expecting results.

b. also included the actual country-level indicators they describe in the Introduction and also Discussion section (e.g., country's level of income inequality, educational democratization, gender inequality) into (if possible) the analyses, or in displaying the results (e.g. GINI coefficient, GDP). In this way, their claims in the Discussion section about why such country differences might exist, could be presented more firmly (backed-up with actual data).

Minor comments:

1. sometimes I miss references to claims made in the Introduction section:

- page 3, line 79: PsyCap can .....high-quality life.

- page 8, line 204: However, ....grow or decline.

2. The authors often use 'etc' after giving examples, which is ambiguous and should be removed (line 104, 177, 220, 251, 254).

3. There is still one claim in the Discussion section that I believe should be formulated more carefully since this cannot be concluded from the results of this cross-sectional study, namely on page 29, line 628-629: This finding also underscores....and vice versa.

Reviewer #2: Thank you for your thorough revisions in response to the reviewer's feedback. While I acknowledge the effort invested, several key issues persist:

1. The justification for introducing the concept and measurement of capital remains unclear. The selection of measurement tools appears arbitrary, for example in excluding occupation from capital measures. This raises concerns about the consistency and validity of the measurements and challenges the characterization of these constructs as "capital."

2. Similarly, the exclusion of ‘hope’ from the PsyCap construct due to data constraints raises questions about the integrity of the composite. Clarification is needed regarding whether it is appropriate to continue labeling it as PsyCap or if it should be considered a composite of other factors. Furthermore, the potential contributions of this concept and measurement to existing literature are not clearly articulated.

3. Additionally, the discussion surrounding the number of Latent Class Analysis (LCA) clusters lacks clarity -- which I have missed in my first-round review. While a 6-cluster model is presented, the marginal decrease in Bayesian Information Criterion (BIC) raises questions about the meaningfulness of these clusters (0.11%). The difference in BIC between 1-cluster and 2-clusters is also negligible (3%), suggesting a lack of significance. It is imperative that the authors provide a robust defense for the meaningfulness of these clusters, tying back to the broader question of the necessity of PsyCap.

7. PLOS authors have the option to publish the peer review history of their article (what does this mean?). If published, this will include your full peer review and any attached files.

Reviewer #1: **Yes: **K. Schelleman-Offermans

Reviewer #2: No

---

## [Author Response · Author response to Decision Letter 1]

30 Apr 2024

Reviewer #1: 

[Authors]: Dear Reviewer #1, we want to explicitly thank you for taking the time to review our revised manuscript with great care. We appreciate your insights and are confident that the manuscript only improved by adjusting the manuscript in accordance with your feedback. 

1) The authors response to comment 3 is not completely satisfying. Ridit scores do not remove country-level effects, but make the measures more comparable over countries. For instance, what is considered high wealth or highly educated can differ between countries. When calculating ridit scores for each measure and per country separately, before placing individuals into a low, middle, or high group, makes measures more comparable over countries. In this case you are not comparing oranges with apples. Calculating ridit scores is therefore especially important when using more objective measures. On the other hand, I do not think the results will change much if the authors would have included ridit scores, since they mostly recoded their measures into quite crude groups (categorized indicators into 3 groups (low-mean-high)) and mostly also did not use objective measures but more subjective measures, where one could expect that these should be comparable over countries. 

However, it might be good to have a statistician have a look, since the sophisticated methods the authors used go beyond my own knowledge.

[Authors]: Thank you for pointing our attention to this important discussion. The discussion about the usage of ridit-scores touches upon a fundamental debate regarding comparing social classes across different countries. This debate revolves around whether we should focus on absolute differences or relative differences. We agree with Reviewer 1 that relative differences matter. To some extent, when comparing personal or household income across countries, relative income comparisons allow for a more accurate assessment of the standard of living within and between countries. Comparing income across countries in a relative way takes into account differences in, for example, purchasing power. A certain income level might afford a higher standard of living in a country with a lower cost of living compared to a country with a higher cost of living. By considering relative income, we could get a more accurate picture of what individuals can afford with their income in different countries. So, ridit scores could provide insights into how income levels in one country compare to those in others, taking into account differences in purchasing power and cost of living between countries.

Yet, for household income (or economic capital in our manuscript), we already using a subjective measure of income: perception of the current household income with answer categories: (1) Living comfortably on present income; (2) Coping on present income; (3) Difficult on present income and (4) Very difficult on present income. The consequence is that we already take country-specific issues into account that could impact the “purchasing power” of an income.

The same goes for psychological capital. People evaluate themselves and their circumstances relative to others. Psychological measures that capture constructs like self-esteem, and resilience are influenced by comparisons with others (see, for instance, social comparison theory). Yet, here too, the different sub-questions that make up the PsyCap scale are subjective questions. For instance, “In general I feel very positive about myself”, with answer categories ranging from 1 (agree strongly) to 5 (disagree strongly). By using these subjective questions, we automatically take country-specific issues into account that could impact how high or low a person might evaluate his or her PsyCap.

The same reasonings can be made for the variable “strong ties” and “weak ties” of social capital, as this is also measured using a subjective measure. 

However, regarding cultural capital, we are using respondents’ own and their parent’s educational attainment based on the ISCED scale. This is measured in an “absolute” manner. Yet, the International Standard Classification of Education (ISCED) exists precisely because we recognize and categorize educational levels in an absolute sense, irrespective of relative differences between countries. Therefore, we are confident in the presentation of our results.

2) Regarding the third aim of the paper, where the authors included the suggested aim of examining country differences in the distribution of social classes, it would have been nice if the authors:

a. included hypotheses into the objectives section of the introduction, about the specific country-level indicators they are expecting results.

[Authors]: Thank you for this suggestion, we agree with Reviewer 1 that including hypothesis will improve the manuscript. We included two hypotheses based on the research of Andersen & Curtis (2012) and Breen (2010):

Our last aim (objective 3) is to examine cross-national differences in social classes distribution. The sizes of social classes within European countries are influenced by macro-level factors, such as a country’s level of economic development, income inequality and educational expansion (Andersen & Curtis 2012; Breen 2010). Countries with less economic development and high-income countries with large income inequality demonstrate more (self-identified) low social classes (Andersen & Curtis 2012). Moreover, educational expansion can lead to a large group of the upper class (with high volume of cultural and economic capital) (Breen 2010). Because there are transformation processes between all types of capital and because countries’ macro-characteristics influence capital volume and composition, it is hypothesized that there will be a greater proportion of social classes with low general capital volume and deprived capital composition in countries with low economic development (such as Ukraine, Kosovo and Albania), in countries with high income inequality (such as Israel, Southern European and Anglo-Saxon countries) and in countries with low educational expansion (such as Southern European countries). A greater proportion of social classes with high general capital volume and rich capital composition will be found in their counterparts (such as Western and Northern European countries).

b. also included the actual country-level indicators they describe in the Introduction and also Discussion section (e.g., country's level of income inequality, educational democratization, gender inequality) into (if possible) the analyses, or in displaying the results (e.g. GINI coefficient, GDP). In this way, their claims in the Discussion section about why such country differences might exist, could be presented more firmly (backed-up with actual data).

Thank you for this suggestion, we have included GDP per capita (in US$), the GINI index and Expenditure on education (% of the GDP), in Europe in 2012 and added these numbers to our hypothesis (see above). These numbers are also added to the results. We have deleted the figure and inserted a Table in which it was easier to include the macro-indicators. The results and discussion have been rewritten to also include the macro-indicators.

 GDP per capita (US$) GINI Educational Expansion1 Low-educated deprived Middle- educated deprived High Psy/soc/eco capital Upward well-off High Psy/soc/cult capital Generationally well-off

Kosovo 3,410 29 n.a. 47 8 15 4 25 2

Ukraine 4,004 24.7 6.44 10 37 8 0 35 10

Albania 4,247 29 3.31 70 1 6 15 3 4

Bulgaria 7,430 36 3.48 35 20 0 2 31 12

Hungary 12,984 30.8 4.14 24 50 0 0 6 20

Poland 13,010 33 4.86 35 1 30 24 1 8

Lithuania 14,367 35.1 4.76 33 23 0 4 31 9

Russian Federation 15,420 40.7 3.79 23 38 12 0 23 3

Estonia 17,403 32.9 4.72 17 22 2 5 39 16

Slovakia 17,498 26.1 3.86 15 57 0 0 9 18

Czechia 19,870 26.1 4.22 1 62 0 0 24 13

Portugal 20,563 36 4.95 54 0 7 33 0 7

Slovenia 22,641 25.6 5.62 17 36 4 0 25 18

Spain 28,322 35.4 4.47 36 1 22 26 0 15

Cyprus 28,910 34.3 5.92 42 11 18 22 0 7

Israel 33,156 41.3 5.59 29 14 26 11 9 11

Italy 35,051 35.2 4.06 44 10 27 7 5 8

France 40,870 33.1 5.46 23 4 51 12 2 9

United Kingdom 42,497 33.1 5.63 22 6 38 20 1 13

Germany 43,855 31.1 4.93 16 0 58 18 2 7

Belgium 44,670 27.5 6.26 25 13 26 17 1 19

Iceland 45,995 26.8 7.58 0 17 39 24 6 13

Finland 47,708 27.1 7.15 0 5 39 6 41 9

Ireland 48,943 33.2 6.162 32 7 37 15 0 8

Netherlands 50,070 27.6 5.41 36 7 22 20 0 15

Sweden 58,037 27.6 7.57 11 15 39 15 7 12

Denmark 58,507 27.8 7.24 6 2 48 17 10 15

Switzerland 85,836 31.6 4.9 10 0 59 20 0 10

Norway 102,175 25.7 7.33 0 10 33 33 6 18

1Government expenditure on education, total (% of GDP); 2 Data from 2009 (data 2012 unavailable); n.a.: not available 

Minor comments:

1. sometimes I miss references to claims made in the Introduction section:

- page 3, line 79: PsyCap can .....high-quality life.

Four References have been added to his sentence. The references added are: 

PsyCap leads to: 

- Good connections (Guo et al. 2020)

- Good jobs (Cenciotti et al. 2017)

- Wealth (Judge & Hurst 2008)

- High quality of life (Santisi et al. 2020)

- page 8, line 204: However, ....grow or decline.

This has been reported in the book “Psychological capital and Beyond” by Luthans et al. (2015). This reference has been added to the manuscript. 

2. The authors often use 'etc' after giving examples, which is ambiguous and should be removed (line 104, 177, 220, 251, 254).

In the reported lines with “etc’” in the sentence, we have rewritten the sentences and deleted the “etc”.

3. There is still one claim in the Discussion section that I believe should be formulated more carefully since this cannot be concluded from the results of this cross-sectional study, namely on page 29, line 628-629: This finding also underscores....and vice versa.

We appreciate the alertness of Reviewer 1. The problem with using cross-sectional data is that you cannot make claims about the direction of the relation. With our study, we cannot say that PsyCap leads to social capital or social capital to PsyCap, you can only say that the two are related, in the moment in time under study. Therefore, the sentence has been changed to: This finding also underscores the importance of the association between social capital and PsyCap.

Reviewer 2: 

Reviewer #2: Thank you for your thorough revisions in response to the reviewer's feedback. While I acknowledge the effort invested, several key issues persist:

1) The justification for introducing the concept and measurement of capital remains unclear. 

Thank you for this comment. We hope we understood it correctly and understand that Reviewer 2 is asking for a justification of PsyCap to be included in the Bourdieusian capital framework. Therefore, we have rewritten the second paragraph of our manuscript. This paragraph is now dedicated to the added value of PsyCap into social inequality research (in a concise manner). The added value of PsyCap, is adding a more “personal” element, free from socialization into the drivers of social inequality. 

The selection of measurement tools appears arbitrary, for example in excluding occupation from capital measures. This raises concerns about the consistency and validity of the measurements and challenges the characterization of these constructs as "capital."

Thank you for this comment. We acknowledge the importance of selecting the correct measurement tools for the different types of capital. The capital types that are included are selected with great care (please see Appendix A, for our answer to comment 1 from Reviewer 2 in the first round of revisions, who made a similar remark). 

Firstly, we have deleted the sentence “we only selected “pure” capitals” from our manuscript as it might reflect a sense of superiority. This is not the purpose of the selected measurement tools. We simple want to test our first hypothesis (the transferability of capitals), which requires items that distinctly reflect only one type of capital. We want to stress that for other research purposes occupational status can be a good indicator to construct social classes.

The exclusion of “occupation” has a specific reason. Occupation reflects both cultural and economic capital (that is also the reason why many class schemes rely only on the characteristics of people’s occupation). Someone’s occupation is frequently a direct result of someone’s education and training, while occupation is also frequently used to reflect someone’s level of income (i.e., economic capital). Although our data include measures for individuals’ occupational status and parent’s occupational status, we decided to not use these items. Parents’ and the respondents’ own occupational status is not a “distinct” measure of for instance economic capital as it also entails cultural capital. 

By using occupational position, we would be unable to test our first aim (the transferability between different capitals). That is the reason why we used “distinct” economic capital variables, such as perception on income and “distinct” cultural capital variables: own’s and parent’s level of education. 

This sentence was added to the methods section: 

“To adequately test the transferability between different capitals, we selected measures that distinctly represent only one form of capital. Therefore, items that could account for two capital dimensions are excluded. For instance, occupational status reflects both cultural capital (as educational attainment often is a prerequisite for access to certain professions) and economic capital (as a job is a primary mean of income and financial security) and is subsequently omitted from our analyses.”

2) Similarly, the exclusion of ‘hope’ from the PsyCap construct due to data constraints raises questions about the integrity of the composite. Clarification is needed regarding whether it is appropriate to continue labeling it as PsyCap or if it should be considered a composite of other factors. Furthermore, the potential contributions of this concept and measurement to existing literature are not clearly articulated.

Thank you for pointing our attention to the limitations in the PsyCap construct. We partly agree with Reviewer 2 that we need to be more careful in labelling our construct as PsyCap. Therefore, we have now more clearly indicated in the manuscript that we measure “a proxy” of PsyCap.

In large cross-national surveys such as the ESS one often needs to work with proxy variables. For PsyCap this means that we have adequate items for self-efficacy, resilience and optimism, but not for hope. What we did in our revised paper, however, is to explicitly mention that our measure of PsyCap is a proxy measure.

The consequence of using a proxy is one of construct validity. We, however, feel that this issue is not too severe because (a) PsyCap is a higher-order construct that captures shared variance between four lower-order constructs. Because three of those constructs are well represented in our proxy measure and because our analyses show that they clearly load on one factor and are internally consistent, we believe that our proxy captures this common core rather well. 

Moreover, (b) Hope consists of three things: agency, pathways, and goal (Snyder & Lopez 2002). Whereas pathways and goals are not well represented in the items, agency is to some extent reflected in the resilience item "How difficult or easy do you find it to deal with important problems that come up in your life". In addition, according to Huppert et al. (2013), optimism also refers to hopefulness, by reflecting its emotional facet. So, hope is somewhat captured in the way optimism and resilience are measured. 

We added this issue to our limitations. We added several sentences in the limitations section: 

“Additionally, the indicator for PsyCap is only a proxy for the underlying theoretical concept and the measurement of the dimension hope was suboptimal. Yet, because (1) PsyCap is a second-order construct, (2) the subscales clearly load on one factor and are internally consistent and (3) hope is somewhat captured in the way optimism and resilience are measured, the inadequate measurement of one indicator should not be problematic.

---

## [Decision Letter · Decision Letter 2]

23 Jun 2024

PONE-D-23-17518R2Psychological capital and social class: A capital approach to understanding positive psychological states and their role in explaining social inequalitiesPLOS ONE

Dear Dr. De Moortel,

Thank you for submitting your manuscript to PLOS ONE. After careful consideration, we feel that it has merit but does not fully meet PLOS ONE’s publication criteria as it currently stands. Therefore, we invite you to submit a revised version of the manuscript that addresses the points raised during the review process.

We look forward to receiving your revised manuscript.

Kind regards,

Fraide Agustin Ganotice, PhD

Academic Editor

PLOS ONE

Journal Requirements:

**Additional Editor Comments:**

Dear Authors,

Thank you for submitting your paper titled “Psychological capital and social class: A capital approach to understanding positive psychological states and their role in explaining social inequalities ” to PLOS ONE. Two reviewers have examined the manuscript for which one recommended minor revision and one recommended acceptance. I also went over the manuscript and agreed with the two reviewers to recommend minor revision. This resubmitted version has been greatly improved and only minor changes are needed. Looking forward to seeing the final version!

Thanks so much.

Respectfully yours,

Fred Ganotice

Reviewers' comments:

Reviewer's Responses to Questions

**Comments to the Author**

1. If the authors have adequately addressed your comments raised in a previous round of review and you feel that this manuscript is now acceptable for publication, you may indicate that here to bypass the “Comments to the Author” section, enter your conflict of interest statement in the “Confidential to Editor” section, and submit your "Accept" recommendation.

Reviewer #1: All comments have been addressed

Reviewer #2: All comments have been addressed

2. Is the manuscript technically sound, and do the data support the conclusions?

Reviewer #1: Yes

Reviewer #2: Partly

3. Has the statistical analysis been performed appropriately and rigorously? 

Reviewer #1: Yes

Reviewer #2: Yes

4. Have the authors made all data underlying the findings in their manuscript fully available?

Reviewer #1: Yes

Reviewer #2: (No Response)

5. Is the manuscript presented in an intelligible fashion and written in standard English?

Reviewer #1: Yes

Reviewer #2: Yes

6. Review Comments to the Author

Reviewer #1: I have read the revised manuscript with great pleasure and I the authors have successfully addressed all my comments. It has become a very nice paper that will make a real contribution to the field.

Reviewer #2: I appreciate the extensive revisions you have made and the effort to address the concerns regarding your measurement of PsyCap. Your detailed response highlights your commitment to improving the clarity and robustness of your study.

One thing I would like to highlight is the PsyCap measurement. I don't think the PsyCap measurement should be justified as stated in the discussion: "because (1) PsyCap is a second-order construct, (2) the subscales clearly load on one factor and are internally consistent, and (3) hope is somewhat captured in the way optimism and resilience are measured, the inadequate measurement of one indicator should not be problematic." (p.32).

This justification overlooks the importance of each component in maintaining the integrity of the PsyCap construct (see the discussion of similarities and differences among four dimensions of PsyCap in Lutherans & Youssef-Morgan 2017). Each of the elements contributes uniquely to the overall construct, and their interrelationships do not justify the exclusion of any single component. The integrity and theoretical foundation of PsyCap rest on the presence of all four factors, and omitting one undermines the validity of the construct.

The impact of not fully measuring PsyCap could potentially skew the study's findings and their implications. Although I understand the limitation of secondary data, this needs to be explicitly stated in the manuscript. If possible, I suggest adding a discussion of the expected outcomes that might result from the inclusion of the hope dimension. For instance, in relation to one of the main findings—that countries where governments allocate large funds to education systems (like Norway, Finland, and Denmark) have classes with lower prevalence of low cultural capital—each social and cultural context can provide different levels of hope for individuals. This variation in hope levels could potentially alter the latent classes and demonstrate different cross-cultural dynamics. By including hope, you could offer a more comprehensive analysis of how different components of PsyCap interact with educational and cultural policies across diverse contexts — for example.

7. PLOS authors have the option to publish the peer review history of their article (what does this mean?). If published, this will include your full peer review and any attached files.

Reviewer #1: **Yes: **Karen Schelleman-Offermans

Reviewer #2: No

---

## [Author Response · Author response to Decision Letter 2]

24 Jul 2024

Reviewer #2: 

[Reviewer #2]: I appreciate the extensive revisions you have made and the effort to address the concerns regarding your measurement of PsyCap. Your detailed response highlights your commitment to improving the clarity and robustness of your study.

[Authors:] We are grateful that Reviewer 2 sees the efforts we made to improve our study. Below, we address the final comment.

[Reviewer #2]: One thing I would like to highlight is the PsyCap measurement. I don't think the PsyCap measurement should be justified as stated in the discussion: "because (1) PsyCap is a second-order construct, (2) the subscales clearly load on one factor and are internally consistent, and (3) hope is somewhat captured in the way optimism and resilience are measured, the inadequate measurement of one indicator should not be problematic." (p.32).

This justification overlooks the importance of each component in maintaining the integrity of the PsyCap construct (see the discussion of similarities and differences among four dimensions of PsyCap in Lutherans & Youssef-Morgan 2017). Each of the elements contributes uniquely to the overall construct, and their interrelationships do not justify the exclusion of any single component. The integrity and theoretical foundation of PsyCap rest on the presence of all four factors, and omitting one undermines the validity of the construct.

The impact of not fully measuring PsyCap could potentially skew the study's findings and their implications. Although I understand the limitation of secondary data, this needs to be explicitly stated in the manuscript. If possible, I suggest adding a discussion of the expected outcomes that might result from the inclusion of the hope dimension. 

For instance, in relation to one of the main findings—that countries where governments allocate large funds to education systems (like Norway, Finland, and Denmark) have classes with lower prevalence of low cultural capital—each social and cultural context can provide different levels of hope for individuals. 

This variation in hope levels could potentially alter the latent classes and demonstrate different cross-cultural dynamics. By including hope, you could offer a more comprehensive analysis of how different components of PsyCap interact with educational and cultural policies across diverse contexts — for example.

[Authors:] Thank you for pointing our attention to the paper of Luthans & Youssef-Morgan [1] in the Annual Review of Organisational Psychology and Organisational Behavior. After careful reading of this paper, we altered our discussion. We now make an in-depth discussion of how our results can be skewed due to the lack of a direct measure of “hope” in our PsyCap construct. Moreover, we have deleted our original justification that Reviewer 2 mentions in their review, both in the discussion and in the methods section.

We added the following sentences to our discussion:

The lack of “hope” in our PsyCap construct can potentially skew our results as PsyCap is a second-order construct based on shared commonalities of first-order psychological resources (hope, resilience, efficacy, and optimism), but also based on their unique characteristics [1]. The discriminant validity of these constructs has been empirically established [2]. Unfortunately, the ESS data do not include a proxy for the ‘hope’ element of PsyCap. Given that we used PysCap to define clusters, our analysis may have missed a social class cluster distinctly characterized by waypower, which is unique to hope. This “overlooked” social class might also have a specific cross-country distribution as the level of hope varies between countries and may be related to policies like government spending on education and social services [3]. Whether such a cluster exists and how it is distributed over countries is an empirical question that we cannot answer with the available data. There are, however, certain indications that suggest that the component “hope” does not distinctly dominate profiles within populations. Indeed, studies based on person-centered latent profile analyses of the PsyCap components show a gradient in the profiles (ranging from low, to moderate to rich (general) PsyCap) [4–6] or in other cases the other PsyCap components (optimism or resilience) seem to dominate the profiles [7,8]. Our research shows that there is much to be gained by including PsyCap in comparative research and recommends that a more comprehensive measure for PsyCap is included in large-scale survey projects like the European Social Survey. 

References

1. Luthans F, Youssef-Morgan CM. Psychological Capital: An Evidence-Based Positive Approach. Annu Rev Organ Psychol Organ Behav. 2017;4: 339–366. doi:10.1146/annurev-orgpsych-032516-113324

2. Luthans F, Youssef CM. Emerging Positive Organizational Behavior. J Manage. 2007;33: 321–349. doi:10.1177/0149206307300814

3. Krafft AM, Guse T, Slezackova A, editors. Hope across cultures: Lessons from the International Hope Barometer. Cham: Springer International Publishing; 2023. doi:10.1007/978-3-031-24412-4

4. Song L, Zhou Y, Zheng J, Wang Y, Li H, Feng X. Psychological capital of Chinese employees: Investigating its measurement and latent profiles. J Pacific Rim Psychol. 2024;18. doi:10.1177/18344909241254497

5. Gao Y, Yue Y, Li X. The relationship between psychological capital and work engagement of kindergarten teachers: A latent profile analysis. Front Psychol. 2023;14. doi:10.3389/fpsyg.2023.1084836

6. Teng M, Wang J, Jin M, Yuan Z, He H, Wang S, et al. Psychological capital among clinical nurses: A latent profile analysis. Int Nurs Rev. 2023. doi:10.1111/inr.12918

7. Bouckenooghe D, De Clercq D, Raja U. A person-centered, latent profile analysis of psychological capital. Aust J Manag. 2019;44: 91–108. doi:10.1177/0312896218775153

8. Geremias RL, Lopes MP, Soares AE. Psychological Capital Profiles and Their Relationship With Internal Learning in Teams of Undergraduate Students. Front Psychol. 2022;13. doi:10.3389/fpsyg.2022.776839

9. Platania S, Paolillo A. Validation and measurement invariance of the Compound PsyCap Scale (CPC-12): a short universal measure of psychological capital. An Psicol. 2022;38: 63–75. doi:10.6018/analesps.449651

10. Luthans F, Youssef-Morgan CM, Avolio BJ. Psychological Capital and Beyond. Oxford University Press; 2015.

---

## [Decision Letter · Decision Letter 3]

23 Aug 2024

Psychological capital and social class: A capital approach to understanding positive psychological states and their role in explaining social inequalities

PONE-D-23-17518R3

Dear Dr. De Moortel,

We’re pleased to inform you that your manuscript has been judged scientifically suitable for publication and will be formally accepted for publication once it meets all outstanding technical requirements.

Kind regards,

Fraide Agustin Ganotice, PhD

Academic Editor

PLOS ONE

Additional Editor Comments (optional):

Dear Authors,

Thank you for submitting your paper titled “Psychological capital and social class: A capital approach to understanding positive psychological states and their role in explaining social inequalities ” to PLOS ONE. Two reviewers have examined the manuscript andrecommended acceptance. I also went over the manuscript and agreed with the two reviewers to recommend acceptance.

Thanks so much.

Respectfully yours,

Fred Ganotice

Reviewers' comments:

Reviewer's Responses to Questions

**Comments to the Author**

1. If the authors have adequately addressed your comments raised in a previous round of review and you feel that this manuscript is now acceptable for publication, you may indicate that here to bypass the “Comments to the Author” section, enter your conflict of interest statement in the “Confidential to Editor” section, and submit your "Accept" recommendation.

Reviewer #2: All comments have been addressed

2. Is the manuscript technically sound, and do the data support the conclusions?

Reviewer #2: Partly

3. Has the statistical analysis been performed appropriately and rigorously? 

Reviewer #2: Yes

4. Have the authors made all data underlying the findings in their manuscript fully available?

Reviewer #2: (No Response)

5. Is the manuscript presented in an intelligible fashion and written in standard English?

Reviewer #2: Yes

6. Review Comments to the Author

Reviewer #2: I have reviewed the revised manuscript. In the revised manuscript, all comments have been addressed.

7. PLOS authors have the option to publish the peer review history of their article (what does this mean?). If published, this will include your full peer review and any attached files.

Reviewer #2: No

---

## [Editor Report · Acceptance letter]

29 Aug 2024

PONE-D-23-17518R3 

PLOS ONE

Dear Dr. De Moortel, 

I'm pleased to inform you that your manuscript has been deemed suitable for publication in PLOS ONE. Congratulations! Your manuscript is now being handed over to our production team.

Kind regards, 

on behalf of

Dr. Fraide Agustin Ganotice 

Academic Editor

PLOS ONE